# Transport to the Slaughterhouse Affects the *Salmonella* Shedding and Modifies the Fecal Microbiota of Finishing Pigs

**DOI:** 10.3390/ani10040676

**Published:** 2020-04-13

**Authors:** Francesca Romana Massacci, Alessandra Morelli, Lucilla Cucco, Adrien Castinel, Roberta Ortenzi, Silvia Tofani, Giovanni Pezzotti, Jordi Estellé, Marta Paniccià, Chiara Francesca Magistrali

**Affiliations:** 1Istituto Zooprofilattico Sperimentale dell’Umbria e delle Marche ‘Togo Rosati’, 06124 Perugia, Italy; a.morelli@izsum.it (A.M.); l.cucco@izsum.it (L.C.); r.ortenzi@izsum.it (R.O.); silvia.tofani-esterno@izslt.it (S.T.); g.pezzotti@izsum.it (G.P.); m.paniccia@izsum.it (M.P.); c.magistrali@izsum.it (C.F.M.); 2Department of Agricultural and Food Sciences, University of Bologna, 40127 Bologna, Italy; 3GABI, INRAE, AgroParisTech, Université Paris-Saclay, 78352 Jouy-en-Josas, France ; Jordi.Estelle@inra.fr; 4GeT-PlaGe, Genotoul, INRAE US1426, 31320 Castanet-Tolosan CEDEX, France; adrien.castinel@inra.fr; 5Istituto Zooprofilattico Sperimentale del Lazio e della Toscana ‘M. Aleandri’, 00178 Roma, Italy

**Keywords:** swine, infection, *Salmonella*, intestinal composition

## Abstract

**Simple Summary:**

*Salmonella* is one of the most important pathogens responsible for food poisoning in humans and pork is recognized as one of the major sources for human salmonellosis. Pigs can become infected with *Salmonella* on the farm or in the very last phases of the production chain that include transport, lairage, and slaughter. The transport of animals from the farm to the slaughterhouse plays an important role in the transmission of this pathogen from pigs to pigs. The aim of our study was to investigate if the transport from the farm to the slaughterhouse increases the load of *Salmonella* in feces and to determine a modification of the fecal microbiota in pigs. Our study showed that the load of *Salmonella* increases after transport, confirming that this phase of the production is a critical point for the control of *Salmonella* contamination. The fecal microbiota composition was modified in *Salmonella-positive* animals after transport, in accordance with what is already published in scientific literature. In this stage, a natural *Salmonella* infection causes a severe modification of the fecal microbiota which is similar to the one observed in studies carried out in experimental facilities.

**Abstract:**

Contaminated pork is a significant source of foodborne Salmonellosis. Pork is contaminated at the slaughterhouse and the intestinal content is the predominant source of *Salmonella* for carcass contamination. The prevalence of *Salmonella*-positive pigs increases significantly when the time of transport to the slaughterhouse is longer than two hours. The hypothesis behind this study is that transport to the slaughterhouse increases the load of *Salmonella* in feces and determines a shift of the fecal microbiota in finishing pigs. Fecal samples were collected in a pig herd positive for *Salmonella* spp., the day before the transport and at the slaughterhouse. *Salmonella* loads were estimated by the most probable number (MPN) technique, according to the ISO/TS 6579-2:2012/A1. Moreover, the fecal bacteria composition was assessed by sequencing the V3-V4 hypervariable regions of the 16S rRNA gene. Our study showed that the load of *Salmonella* increases after transport, confirming that this phase of the production chain is a critical point for the control of *Salmonella* contamination. A lower richness and an increased beta-diversity characterized the fecal microbiota composition of *Salmonella*-positive animals after transport. In this stage, a natural *Salmonella* infection causes a disruption of the fecal microbiota as observed in challenge studies.

## 1. Introduction

Salmonellosis is one of the most important foodborne diseases in the EU [1], with 91,857 confirmed cases of salmonellosis in humans reported in 2018. Pork is recognized as one of the major sources for human salmonellosis [1]. Pigs can become infected with *Salmonella* on the farm or in the very last phases of the production chain that includes transport, lairage, and slaughter. The transport of animals from the farm to the slaughterhouse plays an important role in the transmission of this pathogen from pigs to pigs, resulting in an increased risk for the contamination of carcasses [2]. Indeed, the prevalence of *Salmonella*-positive pigs increases significantly when the time of transport to the slaughterhouse is longer than two hours [3]. According to Berends et al., 2–6 h of combined transport and lairage could double the number of animals excreting *Salmonella* [4]. Several determinants may contribute to this phenomenon, which includes stress associated with transport, feed withdrawal, handling, mixing with other pigs, high stocking densities, and changes in the environment, including temperature [2]. Moreover, *Salmonella* infection may be acquired during transport through contact with contaminated trucks or pen floors at lairage [5]. At slaughter, the intestinal content and the feces of carrier pigs are the predominant sources of *Salmonella* for carcass contamination [6]. In a previous paper, we provided evidence that the risk of carcass contamination is higher when the intestinal load of *Salmonella* in pigs at slaughter is ≥10^3^ CFU/g [7]. 

Most studies described a disruption of the porcine gut microbiota as a consequence of *Salmonella* infection [8,9,10]. A higher abundance of the family *Ruminococcaceae* in *Salmonella-negative* or low-shedder pigs and depletion of *Prevotella* in high-shedder pigs early after the infection was reported [10,11,12]. However, the majority of these investigations were carried out in challenge studies, which may not reflect the conditions occurring during a natural infection. Moreover, to the best of our knowledge, no data are available about the modification of the fecal microbiota after transport to slaughter, which represents a critical control point for *Salmonella* contamination in the pork production chain. 

The hypothesis behind this study is that transport longer than two hours to the slaughterhouse could increase the load of *Salmonella* in feces. Furthermore, we explore the effect of transport and *Salmonella* shedding on the fecal microbiota in pigs at slaughter.

## 2. Materials and Methods 

### 2.1. Animal Design

This study involved 50 fattening pigs from a farm located in the Marche region, Central Italy. The number of pigs was estimated with 10 percent expected prevalence 95% CIs and 5 percent precision, in accordance with a previous study [7]. The final number of pigs was calculated taking into account the number of animals delivered to slaughter. This study was performed according to the Italian and European regulations regarding the protection of animals used for scientific purposes (Directive 2010/63/EU, D.L. 26/2014). The herd was positive for *Salmonella* spp. following a pilot study. The transport time to the slaughterhouse lasted approximately six hours. Fecal samples were collected on the farm the day before the transport and at the slaughterhouse. Fecal samples were stored at +4 °C and processed within 12 h for the estimation of the *Salmonella* load. The aliquots for the intestinal microbiota study were directly frozen in liquid nitrogen just after sampling and further stored at −80 °C until use. 

### 2.2. Microbiological Culture 

The microbiological analysis of fecal contents was carried out using a miniaturized most probable number (MPN) technique, according to the ISO/TS 6579-2:2012/A1 [13] protocol, as already described [7]. This technique provides an estimate of the *Salmonella* spp. load, using the most probable number (MPN) method. Briefly, 5 g of the fecal contents were diluted 10-fold in buffered peptone water (BPW) (Oxoid Ltd., UK). This initial suspension was then used to perform 12 serial dilutions in a 1:5 ratio, carried out by systematically transferring an aliquot of 0.5 mL of each successive dilution into 2 mL of BPW. Each dilution was then incubated and processed as described in the procedure. The MPN values and their 95% CI were calculated using the MPN calculator, available on the website http://standards.iso.org/iso/ts/6579/-2. Isolates of *Salmonella* spp. from positive samples were serologically identified (*Salmonella* sera Statens Serum Institut, Denmark) according to the White-Kauffman-Le Minor Scheme [14].

### 2.3. Fecal DNA Extraction and 16S rRNA Gene Sequencing

Genomic DNA of each fecal sample was extracted using the Qiagen QIAamp DNA stool kit, following the modified protocol of Dore et al. [15]. A NanoDrop spectrophotometer (Thermo Scientific, USA) was used to assess the quality of the DNA extracts. Microbial profiling was performed via the high-throughput sequencing of the V3-V4 hypervariable region of the 16S rRNA gene (2 × 250 bp paired-end reads) using an Illumina MiSeq Sequencer (Illumina, USA). We employed the standard Illumina protocol and the primers *PCR1F_343* (5′-CTTTCCCTACACGACGCTCTTCCGATCTACGGRAGGCAGCAG-3′) and *PCR1R_784* (5′-GGAGTTCAGACGTGTGCTCTTCCGATCTTACCAGGGTATCTAATCCT-3′). Quality control was performed on the resulting FastQ files using FastQC software (https://www.bioinformatics.babraham.ac.uk/projects/fastqc/); the files were then analyzed using QIIME software (v. 1.9.1) [16] and the subsampled open-reference OTU picking process [17]. The Illumina adapters were removed trough the “cutadapt” function [18]. Forward and reverse paired-end sequence reads were collapsed into a single continuous sequence according to the “fastq-join” option of the “join_paired_ends.py” command in QIIME. Therefore, the “split_libraries_fastq.py” command was used to demultiplex and filter (at Phred ≥ Q20) the fastq sequence data [19]. Subsequently, the sequences were clustered into an Operational Taxonomic Unit (OTU) against the GreenGenes database [20] by using the uclust algorithm [21] method at a 97% similarity cutoff. Chimeric sequences were removed through the “parallel_identify_chimeric_seqs.py” function in QIIME and by using the BLAST algorithm against the GreenGenes reference alignment [20]. Singleton OTUs and OTUs representing less than 0.005% of the total number of sequences were removed from the dataset as recommended by the QIIME software authors [22]. Chimeric sequences were identified using the BLAST algorithm and removed from the dataset. Samples with fewer than 10,000 reads after quality control procedures were eliminated, which resulted in eliminating only one sample (pig number 18 sampled on the farm).

### 2.4. Biostatistical Analyses

All statistical analyses were performed in R (v. 3.6.1) [23]. The percentages of positive fecal samples on the farm and at the slaughterhouse were compared using the Mc Nemar χ^2^ test. Additionally, the load of *Salmonella* spp. in the feces collected on the farm and at the slaughterhouse were analyzed by the Wilcoxon test for paired data. 

For the fecal microbiota analysis, our hypothesis was split into four different questions: 

(i) Is the *Salmonella* status associated with a different gut microbiota composition in pigs at the slaughterhouse? We compared fecal samples collected at the slaughterhouse, from Salmonella-negative animals and Salmonella-positive pigs (Test 1); 

(ii) Does *Salmonella* infection during transport affect the fecal microbiota of pigs? We included the animals which became *Salmonella*-positive after transport. We compared fecal samples collected from the same pig on the farm and at the slaughterhouse (Test 2);

(iii) Does the transport to the slaughterhouse affect the gut microbiota of pigs? We included only *Salmonella-negative* animals in this group. We compared fecal samples collected from the same pig on the farm and at the slaughterhouse (Test 3);

(iv) Does the fecal microbial composition on the farm predict the ‘will be’ *Salmonella* status at slaughter? We included fecal samples collected on the farm. We compared the fecal microbiota composition of the pigs which were negative throughout the study with the fecal microbial composition of pigs which became *Salmonella* positive at slaughter (Test 4).

The biome OTU table was imported into R using the Phyloseq package (v. 1.28) [24]. The vegan (v. 2.5-6) package [25] was used to perform rarefaction analyses of the OTUs in each experimental group. Richness and diversity analyses were performed at the OTU level. Alpha diversity and beta diversity were calculated using the Shannon index and Whittaker’s index, respectively. Precisely, Shannon’s α-diversity weighs both microbial community richness (number of different species) and evenness (equitability). The homogeneity of dispersions of microbiota composition between samples was tested through Whittaker’s index using the multivariate analyses of the homogeneity of group dispersion (the “betadisper” function of the Vegan R package), that is, the distance of individual samples from the centroid of their experimental group. Richness was defined as the total number of OTUs present in each sample. Alpha diversity, beta diversity, and log-transformed richness were then analyzed using ANOVA (R’s ‘aov’ function); *post-hoc* comparisons were performed with Tukey’s Honest Significant Differences (HSD) test. 

The vegan package was also used to perform the non-metric multidimensional scaling (NMDS) by using Bray-Curtis dissimilarity values using the ‘metaMDS’ function, which allowed assessing differences in the overall composition of fecal microbiota among samples. The NMDS plot was taking into account a stress threshold of 0.1, which reflects well the ordination summarizing the observed distances among the samples. The number of axes or dimensions tested in the NMDS analysis was two. The ‘env_fit’ function was used to evaluate the statistical significance of the study variables within the NMDS ordination space. Permutational multivariate analyses of variance (PERMANOVA) were performed using the ‘adonis’ function. The significance threshold was set at *p* < 0.05.

OTU differential abundance testing was carried out with the metagenomeSeq package (v. 1.26.3) [26] by using the raw OTU counts before rarefaction. OTU counts were normalized using the cumulative sum scaling method, and a zero-inflated Gaussian distribution mixture model (‘fitZig’ function) was employed to assess differences in relative OTU abundance. The significance level was set to a false discovery rate (FDR) lower than 0.05. The model for the OTU differential abundance test included the *Salmonella* status of animals as a factor.

To characterize a set of microbes consistently present in the fecal microbiota of pigs, a detection threshold of 0.001% and a prevalence threshold of 99.9% at the family level (e.g., a given bacterial family must be present in 99.9% of individuals with a relative abundance of at least 0.001%) was employed, using the microbiome R package [27]. 

## 3. Results

### 3.1. Microbiological Culture

The proportion of pigs positive for *Salmonella* did not significantly vary between before and after transport time points: five pigs out of 49 (10.2%, CI95%: 3.8%–23%) were positive for *Salmonella* spp. on the farm, while 13/49 (26.5% CI95%: 15%–41.3%) were positive at the slaughterhouse (Mc Nemar’s chi-square test; *p* = 0.12). Overall, twenty-eight pigs (57.1% CI95%: 42.3%–70.9%) were positive for *Salmonella* on at least one sampling occasion. The detected serotypes both on the farm and at the slaughterhouse were *Salmonella* Rissen and the monophasic variant of *Salmonella* Typhimurium (4.12:i:-). The bacterial loads in the feces ranged from 2 log10 MPN/g, to 6 log10 MPN/g, which was the upper detection limit of the test. The *Salmonella* loads in fecal samples increased after transport (Wilcoxon signed-rank test with continuity correction; *p* = 0.013). The results of the microbiological culture are shown in Table 1.

### 3.2. Fecal Microbiota Composition

#### 3.2.1. Fecal Microbiota Sequencing, OTU Identification and Annotation 

After quality control, a mean of 19,530.51 (S.D. = 4010.74) read counts were available for each sample. Sequences from the whole sample set were successfully clustered into 1237 OTUs, and only 1.29% of the OTUs could not be assigned to a given phylum. OTU counts per sample and OTU taxonomical assignments are available in Appendix A. Overall, 475 out of the 1237 OTUs (38.4%) were assigned to a genus. The phyla *Firmicutes* (854/1237) and *Bacteroidetes* (248/1237) represented 69% and 20% of the OTUs, respectively. Within the phylum *Firmicutes*, 96% (823/854) of the OTUs were assigned to the order *Clostridiales*, 46% (379/823) to the family *Ruminococcaceae*, and 20% (161/823) to the family *Lachnospiraceae*. Within the phylum *Bacteroidetes*, 43% (107/248) were assigned to the genus *Prevotella*. Other phyla were also described, but with low percentages (e.g., *Proteobacteria*: 3.8%, *Spirochaetes*: 3.5%, *Tenericutes*: 0.8%, *Fusobacteria*: 0.1%, *Actinobacteria*: 0.6%, *Fibrobacteres*: 0.4 % and *Deferribacteres*: 0.08%; Figure 1). 

To identify the stability of the fecal microbiota, the core microbiota of each animal was investigated in our study at the family level (bacteria shared by 99% of samples with a minimum detection threshold of 0.001%). The fecal core microbiota in our cohort was composed by 15 families, of which 47% (7/15) belonged to the *Firmicutes* and 40% (6/15) to the *Bacteroidetes* phyla. The 86% (6/7) of OTUs belonging to the *Firmicutes* phylum was assigned to the *Clostridiales* order and more precisely, to the *Ruminococcaceae*, *Christensenellaceae*, *Clostridiaceae*, *Streptococcaceae,* and *Lachnospiraceae* families (Figure 1C). The totality of OTUs belonging to the *Bacteroidetes* phylum was assigned to the *Bacteroidales* order and more precisely, to the *Prevotellaceae* and *Bacteroidaceae* families (Appendix A). 

#### 3.2.2. The *Salmonella* Status Is Associated with a Different Gut Microbiota Composition in Pigs at the Slaughterhouse

In Test 1, comparing fecal samples collected at the slaughterhouse from *Salmonella*-negative animals (N = 20) and *Salmonella*-positive pigs (N = 13), the overall composition of the fecal microbiota at the slaughterhouse was different between negative and positive animals (Adonis test, *p* = 0.02). The beta diversity was different according to the *Salmonella* status of pigs (ANOVA test, *p* = 0.028), showing heterogeneity among samples in *Salmonella*-positive animals. In the NMDS plot, the centroids of the two groups appeared separated, resulting in a significant value (envfit test, *p* = 0.001; Figure 2A). The alpha diversity and the observed microbial richness at the OTU level also showed differences among the *Salmonella*-positive and *Salmonella*-negative animals (ANOVA test, *p* = 0.02 and *p* = 0.001, respectively). *Salmonella*-positive animals showed a lower alpha diversity and richness than the negative ones. The status of *Salmonella* in pigs was used in the model of the differential analysis at the OTUs level, describing 12 DA OTUs at the slaughterhouse (Appendix A). *Salmonella*-positive pigs had DA OTUs belonging mainly to *Prevotella*, *Parabacteroides*, and *Butyricimonas* genera compared to *Salmonella*-negative pigs.

#### 3.2.3. *Salmonella* Infection during Transport Affects the Fecal Microbiota in Pigs

Test 2 included the animals which became *Salmonella*-positive after transport, comparing fecal samples collected from the same pig on the farm and at the slaughterhouse (N = 11). In this test, the fecal microbiota was driven by the status of *Salmonella* (Adonis test, *p* = 0.00001). The beta diversity results differed among animals (ANOVA test, *p* = 0.001; Figure 3B). In the NMDS plot, the centroids of the two experimental groups appeared separated, resulting in a significant value (envfit test, *p* = 0.0001; Figure 3A). The alpha diversity and the observed microbial richness at the OTU level were not significantly different between the negative and positive animals (ANOVA test, *p* > 0.05; Figure 3B). We described 326 DA OTUs (Appendix A and Appendix A). Overall, *Prevotella*, *Coprococcus,* and *Parabacteroides* were more abundant in the positive animals at the slaughterhouse, while *Ruminococcus* was more abundant in negative pigs sampled on the farm.

#### 3.2.4. Transport Has an Impact on the Faecal Microbiota Composition

In Test 3, we included only the *Salmonella*-negative animals, comparing fecal samples collected from the same pig on the farm and at the slaughterhouse (N = 16). We found that the overall composition of the fecal microbiota was not affected by the transport (Adonis test, *p* > 0.05). The beta diversity was different between the fecal samples on the farm and at the slaughterhouse (ANOVA test, *p* = 0.005; Appendix A). In the NMDS plot, the centroids of samples collected on the farm and at slaughter appeared separated, resulting in a significant value (envfit test, *p* = 0.04). The alpha diversity at the OTU level was not significantly different and the observed microbial richness showed differences between the groups (ANOVA test, *p* = 0.004; Appendix A). Differential abundance analyses showed 21 DA OTUs belonging mainly to the genus of *Butyricimonas* and *Bacteroides* which were more abundant in the animals sampled on the farm (Appendix A). 

#### 3.2.5. The Fecal Microbial Composition on the Farm Does not Predict the ‘Will Be’ *Salmonella* Status at Slaughter

In Test 4, we included fecal samples collected on the farm, comparing the fecal microbiota composition of the pigs negative (N = 12) throughout the study with the fecal microbial composition of pigs which became *Salmonella*-positive at slaughter (N = 18). We were not able to find any significant difference in the diversity measures on the farm level between the negative pigs and the ones that became positive at the slaughterhouse. The Adonis test, together with the alpha, beta diversity, and the observed microbial richness at the OTU level did not show differences between the groups. In the NMDS plot, the samples of the two groups did not appear separated, resulting in a non-significant value (envfit test, *p* = 0.08). Differential abundance analyses showed 34 DA OTUs (Appendix A). *Succinivibrio*, *Bacteroides* and *Coprococcus* were the DA OTUs described as more abundant in *Salmonella*-positive pigs compared with the *Salmonella*-negative animals.

## 4. Discussion

The present study investigates the effect of the transport from the farm to the slaughterhouse on *Salmonella* shedding in finishing pigs. The study was focused on the interactions among the transport, the prevalence and load of *Salmonella*, and the fecal microbiota composition on the farm and at the slaughterhouse. 

The impact of the transport of finishing pigs from the farm to the slaughterhouse on *Salmonella* shedding is described in the literature. Transport is necessary for pig production systems [2] and it is known that transport to the slaughterhouse longer than two hours stresses animals [3]. This stress has been associated with an increased *Salmonella* shedding from carrier pigs, determining a spread of this pathogen among animals in this production phase [2]. Even though this phenomenon has not been reproduced in all studies [28,29], an increase in *Salmonella* prevalence after the transport has been detected in most of them [29,30,31]. In our study, we observed a 10.2% prevalence of *Salmonella*-positive animals on the farm and 26.5% at slaughter, which is consistent with the data on *Salmonella* prevalence in Europe and Italy [32]. In our cohort, we also described three animals *Salmonella*-positive on the farm but not shedding it at the slaughterhouse, and this finding could be addressed to the low sensitivity of the culture method used for the detection of *Salmonella*. The isolates belonged to *Salmonella* enterica 4,[5],12:i:– (known as the monophasic variant of *S.* Typhimurium) and *Salmonella* Rissen, which are among the most frequently isolated serotypes in European pigs [6,7,33,34]. We were not able to detect new infections from contaminated trucks during transport since the same serotypes were identified on the farm and at slaughter. Interestingly, we showed that the load of *Salmonella* in feces increased from farm to slaughter within the group. Our study was limited to a small group of pigs, therefore this finding should be interpreted with caution. However, if confirmed, our results reinforce the importance of transport in the *Salmonella* contamination of pork. As described in our previous study, the *Salmonella* load is correlated with the contamination of carcasses: therefore, pigs with a high concentration of *Salmonella* in their feces pose a higher risk for the consumer than low-shedders [7]. Consistently, the reduction of *Salmonella* load in the gut has been proposed as one of the most effective strategies to reduce the human risk of *Salmonella* infection [35]. Highly contaminated pigs at slaughter are likely carrier pigs that are experiencing a recrudescence of infection and/or pigs that became infected during transport through contact with *Salmonella* spp. shedders [36,37]. In fact, during periods of stress, carrier pigs can experience a recrudescence of infection, and *Salmonella* counts in their guts and feces can increase [7]. 

It is well-established that *Salmonella* infection affects the gut microbiota of pigs [8,9,11,12]. Overall, we observed that the phyla *Bacteroidetes* and *Firmicutes* were dominant in the fecal microbiota of finishing pigs. These two taxa accounted for 90% of all the sequences obtained, like in prior studies examining the gut microbiota of pigs [8,9,11,38,39,40]. Moreover, the fecal core microbiota we uncovered was consistent with previous studies in pigs [41,42,43,44]. In our study, we found that *Salmonella* infection caused a shift in the fecal microbiota composition of infected pigs, in agreement with existing studies [8,9]. This shift has been achieved in a few hours and this was found either among pigs at the slaughterhouse or in negative animals on the farm that then became *Salmonella*-positive at the slaughterhouse. The composition of the fecal microbiota of *Salmonella*-negative or positive animals is in accordance with the literature [9,12,45]. In fact, we have described an increased abundance of *Prevotella*, *Coprococcus,* and *Paraprevotella* in *Salmonella*-positive animals, which is in accordance with studies carried out in experimental *Salmonella* challenges [9,12,45]. Moreover, the fecal microbiota of negative pigs showed a higher abundance of *Ruminococcaceae* compared to the *Salmonella*-positive pigs, which is also in accordance with the literature [8,9,11]. Interestingly, our results showed that *Salmonella*-positive pigs had higher beta diversity compared with negative animals, meaning that *Salmonella*-positive pigs had more heterogeneous gut microbiota. 

Globally, our study and the cited studies clearly demonstrate that *Salmonella* leads to alterations in the composition of the pig gut microbiota either on the farm or in experimental conditions [8,9,11]. Nevertheless, we did not show a correlation in the fecal microbiota composition on the farm of the ‘will-be’ high-shedder pigs at slaughter. In the absence of *Salmonella* infection, the transport seems to exert an effect on the fecal microbiota, without causing an evident shift of the microbiota composition, such as the one described during the *Salmonella* infection. Moreover, to confirm our different hypotheses, it will be necessary to conduct further research including a bigger cohort of animals, and also to investigate a broader diversity of environmental conditions and production systems affecting the gut microbiota composition. 

In conclusion, our study showed that the load of *Salmonella* increases after transport, confirming that this phase of the production chain is a critical point for the control of *Salmonella* contamination. In this stage, a natural *Salmonella* infection causes a disruption of the fecal microbiota which is similar to the one observed in challenge studies. Strategies aimed at lowering *Salmonella* infection during transport are urgently needed to reduce the burden of this pathogen to human health.

## Figures and Tables

**Figure 1 animals-10-00676-f001:**
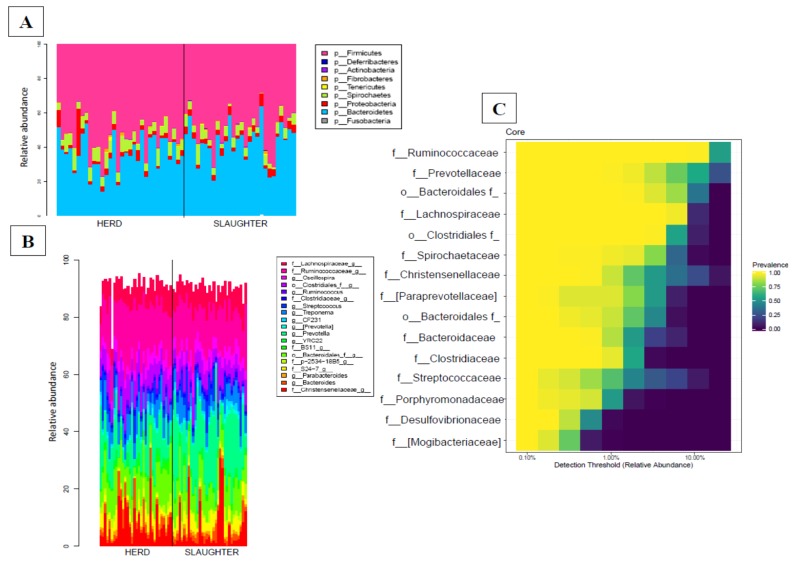
(**A**) Bar plot of the main phyla detected in fecal samples collected on the farm and at the slaughterhouse, respectively; (**B**) Bar plot of the main genera detected in fecal samples collected on the farm and at the slaughterhouse, respectively; (**C**) Heatmap of the fecal core microbiota of pigs. Bacteria were shared by 99% of individuals in our cohort at the family level, with a minimum detection threshold of 0.001%. The x-axis shows the detection threshold of the core microbiota in our cohort.

**Figure 2 animals-10-00676-f002:**
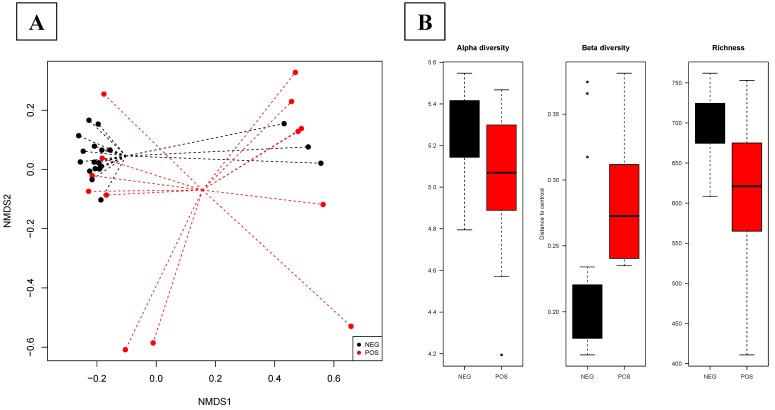
Figure includes fecal samples collected at the slaughterhouse from *Salmonella*-negative animals (N = 20) and *Salmonella*-positive pigs (N = 13) (Test 1). Dissimilarities in fecal microbiota composition represented by the non-metric multidimensional scaling (NMDS) ordination plot, with Bray-Curtis dissimilarity index calculated on unscaled OTU abundances (**A**). The centroids of each group are featured as the group name on the graph (“envfit”; Vegan R package). Samples are colored by the status of *Salmonella*: NEG (negative animals, black) and POS (positive animals; red). Larger filled circles indicate group centroids. (**B**) Box plot graph representation of the alpha diversity (Shannon index), beta diversity (Whittaker’s index), and richness (total number of OTUs present in each sample) using the rarefied OTU table; samples are colored by the status of *Salmonella*: NEG (negative animals, black) and POS (positive animals; red).

**Figure 3 animals-10-00676-f003:**
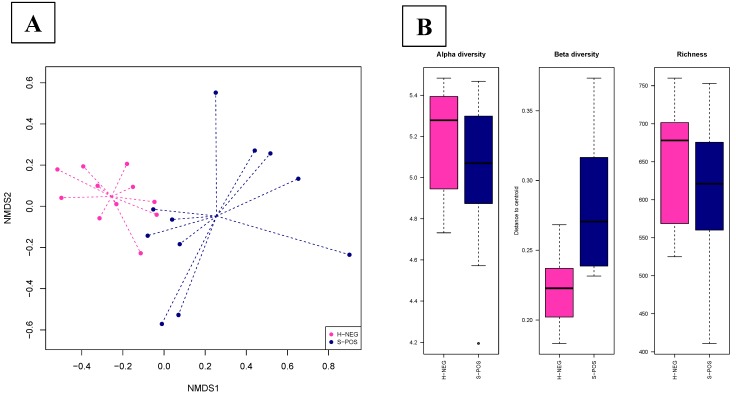
Figure includes animals that became *Salmonella*-positive after transport. We compared fecal samples collected from the same pig on the farm and at the slaughterhouse (N=11) (Test 2). (**A**) Dissimilarities in fecal microbiota composition represented by the non-metric multidimensional scaling (NMDS) ordination plot, with the Bray-Curtis dissimilarity index calculated on unscaled OTU abundances The centroids of each group are featured as the group name on the graph (“envfit”; Vegan R package). Samples are colored by the status of *Salmonella*: H-NEG (negative animals sampled on the farm, pink) and S-POS (positive animals sampled at the slaughterhouse; blue). Larger filled circles indicate group centroids. (**B**) Box plot graph representation of the alpha diversity (Shannon index), beta diversity (Whittaker’s index), and richness (total number of OTUs present in each sample) using the rarefied OTU table; samples are colored by the status of *Salmonella*: H-NEG (negative animals sampled on the farm, pink) and S-POS (positive animals sampled at the slaughterhouse; blue).

**Table 1 animals-10-00676-t001:** Results for *Salmonella* microbiological culture test carried out on the fecal samples collected on the farm or at the slaughterhouse. The MPN/g and relative serotype identified are indicated.

ID	Herd			Slaughterhouse		
	Feces	MPN/g	Serotypes	Feces	MPN/g	Serotypes
1	N	0		N	0	
2	N	0		P	32	*S.* Rissen
3	N	0		N	0	
4	N	0		N	0	
5	N	0		N	0	
6	N	0		N	0	
8	N	0		N	0	
9	N	0		N	0	
10	N	0		N	0	
11	N	0		N	0	
12	N	0		P	78	*S.* 4,12:i:-
13	N	0		N	0	
14	N	0		N	0	
15	N	0		N	0	
16	N	0		N	0	
17	N	0		N	0	
18	N	0		N	0	
19	N	0		P	77	*S.* 4,12:i:-
20	N	0		N	0	
21	N	0		P	74	*S.* 4,12:i:-
22	P	6	*S.* Rissen	N	0	
23	N	0		P	5900	*S.* Rissen
24	N	0		N	0	
25	P	32	*S.* Rissen	P	4000	*S.* 4,12:i:-
26	N	0		P	120	*S.* 4,12:i:-
27	N	0		N	0	
28	N	0		P	51	*S.* 4,12:i:-
29	N	0		N	0	
30	N	0		N	0	
31	N	0		N	0	
32	N	0		N	0	
33	N	0		N	0	
34	N	0		N	0	
35	N	0		N	0	
36	N	0		P	33	*S.* Rissen
38	N	0		N	51	
40	N	0		N	0	
42	N	0		N	0	
43	N	0		N	0	
44	P	100	*S.* Rissen	N	0	
46	P	3100	*S.* 4,12:i:-	P	25000000	*S.* 4,12:i:-
47	P	420	*S.* 4,12:i:-	N	0	
48	N	0		P	160	*S.* 4,12:i:-
49	N	0		N	0	
53	N	0		N	0	
54	N	0		N	0	
55	N	0		P	84	*S.* 4,12:i:-
57	N	0		P	1100	*S.* 4,12:i:-
59	N	6		N	0	

N = negative; P = positive.

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
