# Peer review of "Transport to the Slaughterhouse Affects the Salmonella Shedding and Modifies the Fecal Microbiota of Finishing Pigs"

_animals, 2020, doi:10.3390/ani10040676_

Round 1

Reviewer 1 Report

The manuscript is reporting the relationship between Salmonella and feral microbiome among 50 pigs sourced form one specific farm. The findings will add to the existing knowledge in this area. However, I have the following  concerns:

  • The researchers have tested only 50 pigs from one farm, which I don’t think it is a sufficient sample size to make a relatively accurate conclusion about the impact of transportation on Salmonella shedding and salmonella load (MPN). In fact, I don’t think it should be considered as an objective of this study.
  • Further, the small sample size used to evaluate the four hypothesis explained in lines 114-127 is a concern. I would expect this limitation is discussed and acknowledged in discussion section.
  • The relationship between Salmonella and microbiome is not a univaraiable phenomenon. In fact, other factors such as age, gender, genetics, antibiotic treatment, and other environmental factors may contribute to diversity in fecal bacterial population. Although, the studied pigs in the study originated from one farm, it would be helpful if those information could be presented and considered in data analysis if possible. This also needs to be addressed in discussion section.
  • Another limitation of this study is that the pigs were tested only once on farm, and it is possible that some pigs were not shedding Salmonella at the time of sampling but they were in fact Salmonella positive. Also, the culturing method will have some false negative, This limitation should be discussed.
  • The researchers have tested fecal samples for MPN. I would like to see whether there is any relationship between the diversity in bacterial population and Salmonella MPN?
  • Higher diversity in Salmonella positive pigs should be discussed more thoroughly because the higher diversity is usually an indication of healthier gut.

Line 71: Please state the study objectives clearly.

Line 114-127: It will be helpful if you could use a chart to show the four different test. In fact, here you are reporting the results.  At this stage, the reader do not know the Salmonella status in pigs on farm and at slaughter. My suggestion is that do not present the number here, but report it in the results section.

Line 124, and elsewhere: “‘will be”? it’s not clear what it means.  You may delete it

Line 158: “Overall, twenty-eight pigs (57.1% CI95%: 42.3% -70.9%) were positive for Salmonella…”

This does not agree with the number presented in table 1. Please clarify.

Table 1: Shouldn’t it be 26 positive pigs?

How do you explain the three pigs that were pos on farm but tested neg at slaughter?

Author Response

Reviewer 1

The manuscript is reporting the relationship between Salmonella and feral microbiome among 50 pigs sourced form one specific farm. The findings will add to the existing knowledge in this area. However, I have the following concerns:

The researchers have tested only 50 pigs from one farm, which I don’t think it is a sufficient sample size to make a relatively accurate conclusion about the impact of transportation on Salmonella shedding and salmonella load (MPN). In fact, I don’t think it should be considered as an objective of this study.

AU: We thank the reviewer for the opportunity to discuss this point.

The number of pigs was calculated to estimate the prevalence of pigs carrying a high Salmonella load, using the following parameters:

The expected prevalence of pigs carrying a high Salmonella load their gut (10%) was estimated basing upon a previous study, which has been included in the cited references (Pesciaroli et al., 2017).  The other parameters were a precision of 10% and a 95% confidence. The final number of pigs was calculated taking into account the number of animals delivered to slaughter (lines 74-77).

Our study was longitudinal and aimed at establish a temporal association between transport and infection in the same pig. Longitudinal studies are usually carried out on a limited number of batches and farms and they are not designed to be representative of the pig population. We have chosen a typical farm and data about Salmonella prevalence and serotypes are in line with those described in the literature in Italy. However, we agree that variation in Salmonella prevalence may occur in different batches and farms and this may have had an impact on our findings. Therefore, we included a statement about this limit in the discussion (lines 303-305)

Further, the small sample size used to evaluate the four hypothesis explained in lines 114-127 is a concern. I would expect this limitation is discussed and acknowledged in discussion section.

AU: we discussed the limitation in lines 328-331, as suggested.

The relationship between Salmonella and microbiome is not a univaraiable phenomenon. In fact, other factors such as age, gender, genetics, antibiotic treatment, and other environmental factors may contribute to diversity in fecal bacterial population. Although, the studied pigs in the study originated from one farm, it would be helpful if those information could be presented and considered in data analysis if possible. This also needs to be addressed in discussion section.

AU: we thank the reviewer for the comment. We sampled finishing pigs and, accordingly to the Italian intensive production system, they were around 9-10 months old. Moreover, in intensive swine herds, farmers breed commercial hybrid pigs and antibiotics are not administered within the period just before the slaughter in order to avoid the presence of any antibiotic residues in pork meat. As regards to possible environmental factors affecting the gut microflora, we decided to sample animals belonging to one herd only in order to minimize the impact of the environmental pressure. So according to our explanation, we are not able to address any comparison taking into account the cited factors. However, we discussed this as a limit in lines 328-331

Another limitation of this study is that the pigs were tested only once on farm, and it is possible that some pigs were not shedding Salmonella at the time of sampling but they were in fact Salmonella positive. Also, the culturing method will have some false negative, This limitation should be discussed.

AU: we totally agreed with the reviewer. We discussed the limitation in lines 293-295, explaining that the adopted methodology could have described some false negative results. However, we also sampled the ileo-cecal lymph nodes of animals (results not included in the article) and among the three animals resulted positive on farm but negative at the slaughterhouse we had only one animal tested positive for Salmonella in lymph nodes and other two animals were Salmonella negative either in fecal and in lymph node samples collected at the slaughterhouse.

The researchers have tested fecal samples for MPN. I would like to see whether there is any relationship between the diversity in bacterial population and Salmonella MPN?

AU: we thank the reviewer for the advice. However, we had already carried out this analysis, comparing the low-shedder and vs. high shedder animals according to the MPN results. Since we compared few animals, we did not find any correlation due to the low power of the study. However, this is a pilot study and it could be interesting to confirm our result in a bigger cohort of animals.

Higher diversity in Salmonella positive pigs should be discussed more thoroughly because the higher diversity is usually an indication of healthier gut.

AU: we thanks the reviewer for his note. However, we did not described a lower alpha diversity or species richness in Salmonella positive compared to the negative animals. We described an higher beta diversity in Salmonella positive animals compared the negative animals, meaning that Salmonella positive pigs had more heterogeneous gut microbiota. These findings are coherent with the existing literature, in which a high alpha diversity/richness and a low beta diversity characterize a health status of the gut showing a stable and balanced gut microbiota. Contrarily, a low alpha diversity/richness and an high beta diversity are usually described in animals with enteric diseases.

Line 71: Please state the study objectives clearly.

AU: modified

Line 114-127: It will be helpful if you could use a chart to show the four different test. In fact, here you are reporting the results.  At this stage, the reader do not know the Salmonella status in pigs on farm and at slaughter. My suggestion is that do not present the number here, but report it in the results section.

AU: thanks for the advice. We remove the numbers of pigs tested in each comparison from the materials and methods section and we reported it in the result section as suggested.

Line 124, and elsewhere: “will be”? it’s not clear what it means. You may delete it

AU: we thanks the reviewer for this note. We referred to will-be animals in test 4, in which we investigated if the faecal microbial composition on farm could predict the ‘will be’ Salmonella status at slaughter. For this question, we included fecal samples collected on farm, comparing the fecal microbiota composition of the pigs, which were negative throughout the study with the fecal microbial composition of pigs, which became Salmonella positive at slaughterhouse.

Line 158: “Overall, twenty-eight pigs (57.1% CI95%: 42.3% -70.9%) were positive for Salmonella…” This does not agree with the number presented in table 1. Please clarify.

AU: we totally agreed with the reviewer and we improve the table 1, describing the Salmonella status of all our cohort of animals.

Table 1: Shouldn’t it be 26 positive pigs?

AU: we have included in the table all the microbiological results of our cohort of animals

How do you explain the three pigs that were pos on farm but tested neg at slaughter?

AU: We discussed this finding in lines 293-295, explaining that the adopted methodology could have described some false negative results. However, we also sampled the ileo-cecal lymph nodes of animals (results not included in the article). Among the three animals resulted positive on farm but negative at the slaughterhouse, we described only one animal Salmonella positive in lymph nodes and the other two animals were Salmonella negative either in fecal and in lymph node samples collected at the slaughterhouse.

Reviewer 2 Report

- An ethical statement needs to be included - Figures need to be revised. Change the OTU numbers with taxa name 1C: The numbers on x-axis need to be replaced with taxa name. 3C: Needs clarity. The axes info are not legible. - Line 106: What was the percentage of chimeric sequences? Which database was used to map the OTUs? - Line 133: How was the similarity in composition tested or beta diversity calculated? - Line 142: What was the two-dimensional stress threshold? How many axes were tested? - Line 144:How does the differently abundant OTUs identified? - Recommended to discuss a potential explanation on why positive pigs identified at farm became negative after transport. Line 248: as there was significant separation between pre and post transport samples as indicated by the beta diversity and NMDS observations, transportation seemed to have an effect on fecal flora. So the interpretation as 'limited impact' is not correct.

Author Response

Reviewer 2

- An ethical statement needs to be included

AU: this study was performed according to the Italian and European regulations regarding the protection of animals used for scientific purposes (Directive 2010/63/EU, D.L. 26/2014).

- Figures need to be revised. Change the OTU numbers with taxa name 1C: The numbers on x-axis need to be replaced with taxa name. 3C: Needs clarity. The axes info are not legible.

AU: done

- Line 106: What was the percentage of chimeric sequences? Which database was used to map the OTUs?

AU: the percentage of chimeric sequences was around 2.5% and the sequences were clustered into an OTU using the GreenGenes database. However, a more-detailed description has been included in lines 104-113

- Line 133: How was the similarity in composition tested or beta diversity calculated?

AU: The homogeneity of dispersions of microbiota composition between samples was tested through the Whittaker's index using the multivariate analyses of the homogeneity of group dispersion (the “betadisper” function of the Vegan R package), that is the distance of individual samples from the centroid of their experimental group. Moreover, to assess whether the microbial homogeneity across tested groups was statistically different, we performed an ANOVA using the “aov” function.

- Line 142: What was the two-dimensional stress threshold? How many axes were tested?

AU: the stress threshold was 0.1, which reflects well the ordination summarizing the observed distances among the samples. The number of axes or dimensions tested in the NMDS analysis were two.

 - Line 144: How does the differently abundant OTUs identified?

AU: we used the fitZig function of the metagenomeSeq R package. Moreover, in the model for the OTU differential abundance test was included the Salmonella status of animals as factor (lines 153-158)

- Recommended to discuss a potential explanation on why positive pigs identified at farm became negative after transport.

AU: We discussed this finding in lines 293-295, explaining that the adopted methodology could have described some false negative results. However, we also sampled the ileo-cecal lymph nodes of animals (results not included in the article). Among the three animals resulted positive on farm but negative at the slaughterhouse, we described only one animal Salmonella positive in lymph nodes and the other two animals were Salmonella negative either in fecal and in lymph node samples collected at the slaughterhouse.

- Line 248: as there was significant separation between pre and post transport samples as indicated by the beta diversity and NMDS observations, transportation seemed to have an effect on fecal flora. So the interpretation as 'limited impact' is not correct.

AU: thanks, we have modified the title of the paragraph 3.3.4 and in the discussion section (lines 331-332).

Round 2

Reviewer 1 Report

The authors have responded to my comments, as such I recommend to accept the revised manuscript for publication.  

Author Response

Reviewer 1

The authors have responded to my comments, as such I recommend to accept the revised manuscript for publication. 

AU: we thank the reviewer.

Reviewer 2 Report

The responses to the following previous comments were not included in the revised manuscript. 

  • An ethical statement needs to be included
  • Figures need to be revised. 3C: Needs clarity. The axes' info is not legible. 
  • How was the similarity in composition tested or beta diversity calculated?
  • What was the two-dimensional stress threshold? How many axes were tested?

Please include those.

Author Response

Reviewer 2

The responses to the following previous comments were not included in the revised manuscript.

AU: we apologize for the inconvenient. We have now included our answer in the text.

An ethical statement needs to be included

AU: we have added the ethical statement reported in our previous answer in lines 77-79. “This study was performed according to the Italian and European regulations regarding the protection of animals used for scientific purposes (Directive 2010/63/EU, D.L. 26/2014).”

Figures need to be revised. 3C: Needs clarity. The axes' info is not legible.

AU: we have increased as much we could the size of the text axis. However, since the figure 3C is not one of our main figure explaining our results, we preferred to move the panel 3C in a supplementary figure (Figure S2).

How was the similarity in composition tested or beta diversity calculated?

AU: we have included our previous answer in lines 147-150. “The homogeneity of dispersions of microbiota composition between samples was tested through the Whittaker's index using the multivariate analyses of the homogeneity of group dispersion (the “betadisper” function of the Vegan R package), that is the distance of individual samples from the centroid of their experimental group.”

What was the two-dimensional stress threshold? How many axes were tested?

AU: we have included our previous answer in lines 156-159. “The stress threshold was 0.1, which reflects well the ordination summarizing the observed distances among the samples. The number of axes or dimensions tested in the NMDS analysis were two.”

Please include those.

AU: We thank the reviewer. We have included all the answer in our revised paper. We hope that the paper now meets the requirements of the reviewer.
